# Elemental and Thermochemical Analyses of Materials after Electrical Discharge Machining in Water: Focus on Ni and Zn

**DOI:** 10.3390/ma14123189

**Published:** 2021-06-09

**Authors:** Sergey N. Grigoriev, Marina A. Volosova, Anna A. Okunkova, Sergey V. Fedorov, Khaled Hamdy, Pavel A. Podrabinnik

**Affiliations:** 1Department of High-Efficiency Processing Technologies, Moscow State University of Technology STANKIN, Vadkovsky per. 1, 127055 Moscow, Russia; prof.s.n.grigoriev@gmail.com (S.N.G.); m.volosova@stankin.ru (M.A.V.); sv.fedorov@icloud.com (S.V.F.); eng_khaled2222@mu.edu.eg (K.H.); p.podrabinnik@stankin.ru (P.A.P.); 2Production Engineering and Mechanical Design Department, Faculty of Engineering, Minia University, Minia 61519, Egypt

**Keywords:** electrical erosion, enthalpy, entropy, Ni_5_Zn_21_, removal mechanism, submicrostructure, thermochemistry

## Abstract

The mechanism of the material destruction under discharge pulses and material removal mechanism based on the thermochemical nature of the electrical erosion during electrical discharge machining of conductive materials were researched. The experiments were conducted for two structural materials used in the aerospace industry, namely austenite anticorrosion X10CrNiTi18-10 (12kH18N10T) steel and 2024 (D16) duralumin, machined by a brass tool of 0.25 mm in diameter in a deionized water medium. The optimized wire electrical discharge machining factors, measured discharge gaps (recommended offset is 170–175 µm and 195–199 µm, respectively), X-ray photoelectron spectroscopy for both types of materials are reported. Elemental analysis showed the presence of metallic Zn, CuO, iron oxides, chromium oxides, and 58.07% carbides (precipitation and normal atmospheric contamination) for steel and the presence of metallic Zn, CuO, ZnO, aluminum oxide, and 40.37% carbides (contamination) for duralumin. For the first time, calculating the thermochemistry parameters for reactions of Zn(OH)_2_, ZnO, and NiO formation was produced. The ability of Ni of chrome–nickel steel to interact with Zn of brass electrode was thermochemically proved. The standard enthalpy of the Ni_5_Zn_21_ intermetallic compound formation (erosion dust) Δ*H*^0^_298_ is −225.96 kJ/mol; the entropy of the crystalline phase *S^c^_int_* is 424.64 J/(mol·K).

## 1. Introduction

Electrical discharge machining is a type of machining that consists of changing the geometric parameters, surface quality, and physical properties of the surface of a conductive workpiece under the influence of electric discharges between the workpiece and the tool electrode [1,2,3,4]. Despite the well-known effect of electrical discharges on the surface being machined, the nature of the processes between two electrodes remains unknown.

It should be noted that electrical discharge machining is different from electrochemical machining since the technologies differ in principles. Electrochemical machining is a method for machining electrically conductive materials by the anodic dissolution of the material in an electrolyte under an electric current. The method was proposed in 1911 by the famous Russian chemist E.I. Shpitalsky and modified by V.N. Gusev (W. Gussef), who first proposed conducting machining on the narrow interelectrode gaps (up to tenths of a millimeter) with forced electrolyte pumping [5,6]. The electrochemical discharge machining process is a hybrid machining process having advantages of electrochemical machining and electrical discharge machining and is suitable for nonconductive materials, where electric discharge takes place through the electrolyte and plays a critical role [7]. Unlike these two technologies, electrical discharge machining allows changing the sizes and shapes of electrically conductive materials with high accuracy (±1–2 µm) in a chemically neutral medium of a liquid dielectric (deionized water, hydrocarbons) [8,9]. First developed and patented in 1986 [10], electrical discharge machining techniques can cause electrical destruction of nonconductive materials. However, the developed techniques demonstrate very low efficiency. The maximum achieved depth of the machined holes in a non-oil medium for aluminum-based ceramics (Al_2_O_3_, AlN) does not exceed 700 µm [11] (machining was conducted for a few hours with Ag nanoparticle suspension). The results for Si_3_N_4_ have never been reproduced [12]. ZrO_2_ is easily subjected to electrical discharge machining up to the depth of 5000–8000 µm in an oil medium [13]. However, no one can explain why this technique never worked so efficiently with aluminum-based ceramics or dielectrics. The spectrum of techniques includes modification of workpiece conductivity using the following conductive agents:Conductive nanoparticles introduced into nonconductive matrix [14,15], which have been known since the beginning of the 1980s;Conductive nano- and microparticles introduced into the discharge gap [11,12,16,17,18,19], which were first proposed at the beginning of the 1980s (some of the authors even proposed modifying the interelectrode gap conditions by the introduction of nonconductive particles during machining of conductive materials [20]);Conductive adherent coating of nonconductive workpiece surfaces (auxiliary electrode) [13,14].

In all the proposed techniques, the problem of pulse reinitiation after the completed sublimation of additives from the discharge gap remains.

One way to solve this global problem of efficiency is to research the nature of the erosion process of conductive materials to understand better how the erosion process can be better designed using the chemical properties of chosen electrodes and working medium.

For many years, researchers have declared that the erosion process has thermal nature [21,22] or even mixed thermal and mechanical nature [23] when the physical contact between electrodes is absent, in contrast to the mechanical nature of destruction during milling, turning, and punching operations (physical wear of the tool [24,25,26,27]).

Indeed, the nature of erosional destruction is of a mixed nature, but not thermomechanical; it is thermochemical. Since electrical discharge machining, electrochemical machining, and electrochemical discharge machining are contactless machining methods, there is always an interelectrode gap between the tool and workpiece electrodes, and the physical contact is absent. This allows machining materials despite their hardness or ductility with high precision [5,6,7,28].

The absence of mechanical nature is proved by observation of the even formed surfaces [1,2,3,4,18,21,22] and various works devoted to the research of acoustic oscillations in the full range of spectra [1,29]. Results of research considering erosion products have shown the absence of mechanical contact between electrodes [30,31] and wire oscillations with an amplitude of about ±|9 + *i*| nm (complex number) for steel workpieces and brass wire in water medium excited by working and idle pulses. Thus, the erosion process cannot have any thermomechanical nature since there is no mechanics.

Thermochemical nature is the novelty of this research, proved by X-ray spectroscopy and thermochemical equations of a thermally isolated system (Gibbs energy, entropy, and enthalpy of formation, equilibrium temperature).

The founders of the method, Prof. B.R. Lazarenko and Prof. N.I. Lazarenko [28], declared the thermochemical nature of the phenomena. Since then, it has been raised only once for the combined method of mechanical milling assisted by electric discharge [32] and repeated for electrical discharge machining of dielectric ceramics [33]. A. Calka [32] reported detected erosion products such as amorphous phases, nanocrystalline and quasicrystalline materials, supersaturated solid solutions, reduced minerals, high-surface-area catalysts, and reactive chemicals. The powder initiated solid–solid, solid–liquid, and solid–gas reactions, and a separate materials synthesis and the reaction of workpiece material in a gas atmosphere under discharge pulses were shown. Y. Guo [33] reported on the evidence of the chemical reactions occurring during high-speed wire electrical discharge machining insulating zirconia using assisting electrode technique. Up to now, there have been no other works devoted to the analysis of thermochemical parameters between electrodes in the presence of heat and reactions with the formation of intermetallics.

For the first time, the Lazarenko spouses, who developed the electrical discharge machining method in 1942, mentioned the electrical and chemical nature of the method. Since then, it has been mentioned by only A. Calka and Y. Guo in their articles on chemical transformations occurring on machined surfaces. A. Calka plunged into this issue in sufficient detail from the material science point of view (eutectic, crystalline formation) but only touched on the topic of electrical discharge machining, whileY. Guo casually mentioned the thermochemical nature in the context of the machining of insulating ceramics. None of the authors who have investigated electrical erosion have gone so far as to explain the resulting modifications of the workpiece layers from a thermochemical point of view. There is no explanation of why the assisting alumina powder does not allow intensifying the machining of steel, why sparks and a cloud of black sediment are formed during electrical discharge machining of stainless steel, or why insulating zirconium dioxide can be machined in the hydrocarbon medium using the assistant electrode technique when aluminum oxide and nitride are very laborious to process.

This article presents a new, unique approach to understanding the deep nature of electrical erosion after the discharge channel has formed. It makes it possible to evaluate the erosion products; select them based on the properties of the electrical conductivity of the erosion products; and calculate theoretically the probability of the formation of certain erosion products based on the calculated Gibbs energy, entropy and enthalpy of formation, and equilibrium temperature.

Further understanding of the electrical discharge machining process is no longer possible without a thermochemical analysis of the interaction of the components of the electrodes and the dielectric medium. It will improve the efficiency of electrical discharge machining of conductive and nonconductive materials, give an exhaustive explanation of the observed phenomenon in the discharge channel, avoid combinations of the electrode and working medium materials that reduce the efficiency of electrical discharge machining, and predict the chemical content of the erosion products that assist or hamper machining.

This study is devoted to researching thermochemical interactions (phenomena) between tool electrode, workpiece, and dielectric medium components during wire electrical discharge machining; calculation of thermochemical parameters of erosion products’ formation; and substantiation of chemical reactions by thermochemical approach.

Research on the nature of electrical erosion wear was conducted for two types of structural materials, namely nickel-containing and non-nickel-containing materials, machined with a brass tool electrode in deionized water using spectroscopy and analytical research of the chemical interaction of electrodes and working medium components based on thermochemical parameters such as enthalpy, entropy, Gibbs energy, and equilibrium temperatures. Calculations are provided for the formation of conductive or nonconductive erosion products: Zn(OH)_2_, ZnO, NiO, Ni_5_Zn_21_.

The scientific novelty of the work is in the following:The new data on thermochemical phenomena that occurred between electrodes and deionized water medium in the presence of plasma heat (10,000 °C);Completed thermochemical analyses of chemical interactions at the surface and near-surface layers after electrical discharge machining.

The practical significance of the work is the development of the theoretical method of thermochemical prediction of erosion product chemical content to evaluate the electrical conditions in the discharge gap suitable for conductive and nonconductive materials.

The tasks of the study are the research of the surface topology of two structural materials, namely austenite anticorrosion X10CrNiTi18-10 (12kH18N10T) steel (nickel-contacting) and 2024 (D16) duralumin (non-nickel-containing), after electrical discharge machining with a brass wire tool electrode in a deionized water medium and the thermochemical analysis of chemical reactions between electrodes and working fluid (Figure 1).

## 2. Materials and Methods

### 2.1. Equipment and Machining Factors

A four-axis computer numerical control (CNC) machine Seibu M500S (Seibu Electric & Machinery Co., Ltd., Fukuoka, Japan) (Table 1) was used in the experiments for wire electrical machining in deionized water.

Workpieces were immersed in a working medium for 10 min before machining to avoid dimensional fluctuations of the temperature difference between the environment and the working medium (deionized water). The level of the medium was established at 1–2 mm above workpiece height. The upper wire guide was 2–5 mm above the dielectric medium [34,35]. The wire tool electrode was made of brass (Cu-65%; Zn-35%) Novotec Ultra-Brass wire (Germany) with a diameter *d_w_* of 0.25 mm and a tensile strength of 900 N/mm^2^. The CNC programs were prepared manually; experiment factors were chosen following recommendations of [36,37,38] and are presented in Table 2. The tool electrode had a negative bias, and a workpiece was positive. Flushing was used during experiments to provide better performance and machining stability [39]. The path offset was not taken into account during machining [1,37].

### 2.2. Materials to Be Machined

Two structural materials typical for the aerospace industry and tool production were chosen for the experiments, namely anticorrosion chrome–nickel X10CrNiTi18-10 (12kH18N10T) steel of austenite class (Table 3) and 2024 duralumin (D16, Table 4). The thickness of the samples was 18–20 mm. It should be noted that chromium content provides anticorrosion properties of the steel when nickel is responsible for its austenite class that improves its machinability and extends the exploitation properties. The titanium addition hampers chromium carbides’ formation and forms refractory carbides of titanium in reaction with carbon. This type of chromium–nickel steel dominates the modern rolled metal market [40,41,42,43]. Duralumin is used mainly in a quenched state and is classified as a durable thermohardened construction material for the aerospace industry and unsuitable for welding [44,45].

The specific electrical resistance ρ of the workpieces and wire tool electrode was controlled using a Fischer Sigmascope SMP10 device (Helmut Fischer GmbH, Sindelfingen, Germany) (Table 5) that measures the percentage of the control sample’s electrical conductance made of annealed bronze in the range of 1–112%. The measured values were converted to $\frac{\Omega \cdot {\mathrm{mm}}^{2}}{m}$. The linear thermal expansion coefficient; thermal conductivity; and melting, boiling, and decomposition points are presented in Table 5 [46,47,48,49,50,51].

### 2.3. Characterization of the Samples

The discharge gap and topology were controlled optically using an Olympus BX51M instrument (Ryf AG, Grenchen, Switzerland).

The cross-sections were prepared using Opal 410, Jade 700, and Saphir 300 sample equipment (ATM, Haan, the Netherlands) and standard techniques. An epoxy resin with quartz sand was used as a filler.

Elemental analyses of the machined surfaces were conducted using Thermo Scientific’s K-ALPHA X-ray photoelectron spectrometer (Thermo Fisher Scientific Inc., Bremen, Germany) equipped with an Avantage Data System (version 5.0, Thermo Fisher Scientific Inc., Bremen, Germany).

### 2.4. Thermochemical Analyses

To assess the chemical composition of erosion products, namely the probability of specific chemical reactions in the presence of heat, the authors proposed an estimation technique based on the main thermochemical parameters, namely the enthalpy and entropy of formation, the Gibbs energy, and the equilibrium temperature of the proposed systems [52]:(1)$$\Delta G_{T}^{0}=\Delta H_{298}^{0}-T\cdot \Delta S_{298}^{0}\left[\frac{\mathrm{kJ}}{\mathrm{mol}}\right].$$

It is proposed to focus on the interaction of oxygen with the components of the material of the workpiece and tool electrodes since the electrical discharge machining is carried out in deionized water. Attention is also be paid to the tool electrode components’ behavior and the interaction of the nickel of the anticorrosion steel with the zinc of the tool electrode. For comparison purposes, the other aluminum alloy billet contained no nickel.

The chemical reactions with Zn of brass wire tool electrode and Ni of chrome–nickel steel with water medium can be presented as follows [14,53,54] when NiO is taken as the most likely compound of nickel with oxygen:(2)$${\mathrm{Zn}}^{2+}+2H_{2}O\;\to \;\mathrm{Zn}{\left(\mathrm{OH}\right)}_{2}+H_{2}↑,$$
(3)$${\mathrm{Zn}}^{2+}+H_{2}O\;\to \;\mathrm{ZnO}+H_{2}↑,$$
(4)$${\mathrm{Ni}}^{2+}+H_{2}O\to \mathrm{NiO}+H_{2}↑.$$

Knowing the peculiarity of nickel to form intermetallic compounds with some metals, we will consider the possibility of forming a Ni_5_Zn_21_ intermetallic compound (γ-phase, evidenced by the constancy of the entropy factor) [55] as a more prevalent chemical reaction (reverse corrosion process):(5)$${\mathrm{Ni}}^{2+}+Zn^{2+}\overset{+Zn^{2+}}{\to }\mathrm{NiZn}\overset{+Zn^{2+}}{\to }{\mathrm{NiZn}}_{3}\overset{+Zn^{2+}}{\to }Ni_{5}Zn_{21}{}^{\left(\mathrm{metastable}\;\mathrm{phase}\right)}\overset{+Zn^{2+}}{\to }Ni_{5}Zn_{21},$$
(6)$$5Ni^{2+}+21Zn^{2+}\to Ni_{5}Zn_{21}.$$

A feature of this study is that, although the main thermochemical parameters for the theoretical assessment of reactions can be found in the reference literature, calculating these parameters for an intermetallic compound is an urgent scientific and technical problem.

## 3. Results

### 3.1. Optimization of Machining Factors and Discharge Gap

The electrical discharge machining was conducted based on standard recommendations similarly applicable to both types of materials. The experiments showed that the stabilizing was achieved at 60 and 55 V for operational bias *V*_0_ and 35 and 40 N for wire tension *W_t_* during machining of X10CrNiTi18-10 (12kH18N10T) steel and 2024 (D16) duralumin, respectively. The minimum optically measured discharge gap value Δ corresponds to stable machining mode for both materials (Figure 2). Verification of machining factors provided their optimized values in measuring units of open CNC systems.

### 3.2. Surface Topology (Optical Microscopy)

The topology of the samples after electrical discharge machining of both samples in water is presented in Figure 3. As can be seen, both of the samples have a typical “shagreen” type of surfaces with the differently spaced convexes of sublimated and recast material. The distance between surface convexes is 5–50 µm for the samples made of X10CrNiTi18-10 (12kH18N10T) steel and 10–100 µm for the samples made of 2024 (D16) duralumin. The formed convexes (flakes of secondary material) have an overall size of 7–35 µm for X10CrNiTi18-10 (12kH18N10T) steel and 15–45 µm for 2024 (D16) duralumin. The material surface presents a chaotically sublimated surface of the main material that was heat-affected by the discharge pulses and the convexes with the presence of formed complex secondary structure that has metallic luster for the samples made of X10CrNiTi18-10 (12kH18N10T) steel and black matte and metallic matte sheen for the samples made of 2024 (D16) duralumin. Both surfaces vary significantly from the well-known cut surfaces of the metals (grey glossy and grey matte).

### 3.3. Chemical Nanomodification of the Surface Layer (X-ray Spectroscopy)

The chemical analyses of the sample surface after electrical discharge machining by X-ray photoelectron spectroscopy are presented in Figure 4 and Figure 5. Countable results are shown in Table 5 and Table 6. Both of the surface layers have the presence of the chemical components of the wire tool and complex compounds of secondary order (metastable solid solution and/or eutectics).

Both samples show metallic Zn and copper oxide samples responsible for metallic grey and black sediment on the machined surfaces (Figure 4a,b and Figure 5a,b). Zn is bound in oxide (usually matte white film) (Figure 5a) in the duralumin sample. The quantity analyses confirmed the slight presence of wire tool electrode material (less than 4 at.%). It should be noted that copper (1.4 at.%) found on the X10CrNiTi18-10 (12kH18N10T) steel sample machined surface (Table 6) is 5.6 times more than the 2024 (D16) duralumin sample has (0.25 at.%, Table 7) despite the presence of <5 wt.% of copper in the workpiece material (Table 4).

X10CrNiTi18-10 (12kH18N10T) steel sample also demonstrates slight (less than 3 at.%) presence of Fe_2_O_3_ and Cr_2_O_3_ oxides (Figure 4), while 2024 (D16) duralumin sample (Figure 5) is covered with Al_2_O_3_ film (10.42 at.%) of the primary material. The rest of the workpiece material (66–70 wt.% of Fe, 17–19 wt.% of Cr for X10CrNiTi18-10 (12kH18N10T) steel; 91–95 wt.% of Al for 2024 (D16) duralumin) was sublimated in the interelectrode gap in the form of debris.

C-C, C-O, and C=O (Figure 4f and Figure 5e) bonds correspond to the atmospheric contamination of the samples: ~58 at.% bound carbon for X10CrNiTi18-10 (12kH18N10T) steel (Table 7) and ~40 at.% bound carbon for 2024 (D16) duralumin (Table 7).

Some elements of the workpiece material such as Ni (<11 wt.%), Mn (<2.0 wt.%), Ti (<0.8 wt.%), and Si (<0.8 wt.%) for X10CrNiTi18-10 (12kH18N10T) steel sample and Mn (<0.9 wt.%), Fe (<0.5 wt.%), and Si (<0.5 wt.%) for 2024 (D16) duralumin sample are not presented in X-ray photoelectron spectroscopy data (Table 6 and Table 7). However, other elements such as chlorine (0.89 at.% for steel, 1.35 at.% for duralumin), magnesium (0.22 at.% for steel, 4.38 at.% for duralumin), calcium (2.12 at.% for duralumin), bonded nitrogen (2.21 at.% for steel, 2.59 at.% for aluminum) of water dielectric were deposed.

## 4. Discussion

The measured difference in the discharge gaps between two structural materials is related to their electrical properties. As can be seen from Table 5, they have values of specific electrical resistance differing by a factor of ~14 (0.725 $\frac{\Omega \cdot {\mathrm{mm}}^{2}}{m}$ for steel and 0.052 $\frac{\Omega \cdot {\mathrm{mm}}^{2}}{m}$ for duralumin), which explains the difference in the discharge gap. Since duralumin is more conductive (electrical conductivity is the reciprocal of electrical resistance), the breakdown of the dielectric medium (deionized water) occurs at a greater distance than for stainless steel. According to generally accepted information about the typically recommended spark gaps for materials and electrical resistivity values, the lowest value of the conductive materials is listed for graphite, which also exhibits electrical anisotropy [56], and the highest values for listed for silver [57], metallic carbides (delocalized metal bond *d*-element carbides such as Fe_x_C_y_, WC, TiC, V_x_C_y_, Cr_x_C_y_, and Ni_x_C) [58,59], nitrides [60,61,62], and superconducting intermetallics [63]. The lowest values of the discharge gap correspond to the stable wire electrical discharge machining mode that allows more effective control of part geometry.

The elemental analyses showed that the quantity of zinc in the X10CrNiTi18-10 (12kH18N10T) steel sample is 1.84 times higher than that in the 2024 (D16) duralumin, which can be explained by chemical interaction between Zn of the wire and Ni in steel content. According to the theoretical prediction, it can be concluded that the Ni_5_Zn_21_ intermetallic compound (dark grey sediment) can be formed at the following ratio [55]:(7)$$a^{Ni^{2+}}:b^{Zn^{2+}}=1:2.$$

Let us calculate the entropy of the intermetallic compound in the crystalline form:(8)$$S_{int}^{c}=S_{Ni}^{c}\cdot y_{Ni}+S_{Zn}^{c}\cdot y_{Zn}-R^{\prime }\cdot \left[y_{Ni}\cdot lny_{Ni}+y_{Zn}\cdot lny_{Zn}\right]$$
where $S_{int}^{c}$ is the entropy of the crystalline phase of Ni_5_Zn_21_ intermetallic compound, J/(mol∙K); *S^c^* is the entropy of the crystalline phase of the intermetallic compound components, J/(mol∙K); *y* is the molar fraction of the components, mol; *R* = 8.31446261815324 J/(mol∙K); and the gas constant for a specific gas is $R^{\prime }=\frac{R}{M}$, where *M* is the molar mass of Ni_5_Zn_21_ of 1.667 kg/mol. Then,
(9)$$S_{int}^{c}=29.9\cdot 5.0+41.6\cdot 21.0-\frac{8.314}{1.667}\cdot \left[5.0\cdot ln5.0+21.0\cdot ln21.0\right]=664.10\left[\frac{J}{\mathrm{mol}\cdot K}\right].$$

The entropy of the intermetallic compound formation is determined by the Boltzmann equation taking into account the Stirling theory and the corollary of Hess’s law [64]:(10)$$\Delta S_{int}^{0}=-R^{\prime }\cdot \left[y_{Ni}\cdot lny_{Ni}+y_{Zn}\cdot lny_{Zn}\right]=-\frac{8.314}{1.667}\cdot \left[5.0\cdot ln5.0+21.0\cdot ln21.0\right]=-359.00\;\left[\frac{J}{\mathrm{mol}\cdot K}\right].$$
when entropy is less than zero, reactions occur with a decrease in the degree of chaos (solidification).

The empirical value of electronegativities can be used to estimate the standard heats (enthalpy) of the formation of ionic and metallic compounds with the accuracy of 13.8–20.4% for intermetallics:(11)$$\Delta H_{298}^{0}=-23.066\cdot z\cdot {\left(\varepsilon_{A}-\varepsilon_{B}\right)}^{2}\left[\frac{\mathrm{kkal}}{\mathrm{mol}}\right]$$
or
(12)$$\Delta H_{298}^{0}=-96.623\cdot z\cdot {\left(\varepsilon_{A}-\varepsilon_{B}\right)}^{2}\left[\frac{\mathrm{kJ}}{\mathrm{mol}}\right].$$
where $\Delta H_{298}^{0}$ is the standard enthalpy of formation, kJ/mol; *z* is a number of valence bonds; and *ε_A_* and *ε_B_* are empirical values of electronegativities that are 1.8 for Ni and 1.5 for Zn. The number of metallic bonds in the compound is
(13)z=5+21=26,
then
(14)$$\Delta H_{298}^{0}\left(Ni_{5}Zn_{21}\right)=-4.189\cdot 23.066\cdot 26\cdot {\left(1.8-1.5\right)}^{2}=-225.96\left[\frac{\mathrm{kJ}}{\mathrm{mol}}\right].$$

Gibbs energy will be
(15)$$\Delta G_{T}^{0}\left(Ni_{5}Zn_{21}\right)=\Delta H_{298}^{0}-T\cdot \Delta S_{int}^{0}=-225.96-298\cdot \left(-0.359\right)=-118.98\left[\frac{\mathrm{kJ}}{\mathrm{mol}}\right].$$
when $\Delta G_{T}^{0}<0$, the reaction at normal temperature (298 K) is possible in the forward direction from sublimated ions of metals, and the presence of high heat in the discharge channel (any other source of high heat such as laser beam or bombarding by fast atoms [65,66]) can only accelerate Ni_5_Zn_21_ (γ-phase) solidification. The oxide film formed on the transition metals can prevent the direct reaction under normal conditions. The practice shows that the reaction of Ni with Zn at temperatures above 1000 °C (melting points are 1453 °C for Ni, 419.6 °C for Zn, 1682 °C for NiO, and 1975 °C for ZnO; boiling point of Zn is 906.2 °C (Table 5)) has an explosive character and is accompanied by a series of sparks in the discharge gap [51,67] that can be even acoustically monitored and registered [1]. The thermochemical calculations are presented in Table 8.

The enthalpy of the formation for reactions (2) and (3) is less than zero; thus, these reactions spontaneously take place with the release of heat, in contrast to the reaction with nickel (4). Table 9 shows the calculated entropy, Gibbs energy, and equilibrium temperature for mentioned reactions.

In all reactions, the entropy is more than zero, corresponding to an increase in the degree of chaos (evaporation). That can be confirmed by the appearance of gas bubbles in the discharge gap observed during electrical discharge machining with a brass tool electrode and/or with a nickel-containing workpiece in a water medium. The calculated Gibbs energies for reactions (2) and (3) are almost similar when the entropy of formation is higher for ZnO formation. This means that reaction (3) is more probable than (2) in identical conditions. At the same time, Δ*G*^0^_298_ of reaction (4) is above zero despite having higher entropy of formation than (3). This means that the reaction at normal conditions can only pass in the opposite direction and the formation of nickel oxide occurs with heat absorption only when heated to equilibrium temperatures (672.8 K). Thus, the reaction of Ni and Zn is more likely with the presence of nickel (Figure 6).

## 5. Conclusions

The obtained data demonstrate the potential for nanomodification of the surface sample layer by using different machining processes’ components and controlling it by choosing the working fluid and material of the electrode tool. Such modifications can promote the formation of a fragile sublayer for easier mechanical or electrochemical removal for the products where the recast layer is undesired or, on the contrary, strengthen the sublayer to improve wear resistance of the working surfaces of the product.

The nickel present in stainless steel improves the electrical conditions in the interelectrode gap, but not enough to be comparable to the more electrically conductive material, duralumin. Further research will be devoted to the elemental and thermochemical study of two materials with similar compositions and different nickel contents when treated with a brass electrode in water or hydrocarbons.

The thermochemical analyses of the chemical reactions that occurred during solidification of sublimated material components showed that the formation of ZnO is more likely than that of Zn(OH)_2_ in the presence of Zn of the brass tool electrode and that the formation of Ni_5_Zn_21_ is more likely than that of NiO in the presence of Zn of the brass tool and Ni-containing workpiece (chrome–nickel stainless steel of austenite class). The calculated standard enthalpy of Ni_5_Zn_21_ formation is –225.96 $\left[\frac{\mathrm{kJ}}{\mathrm{mol}}\right]$, the entropy of the crystalline phase is 424.64 $\left[\frac{J}{\mathrm{mol}\cdot K}\right]$, and the equilibrium temperature is 629.42 K.

The obtained data have a fundamental character as the developed method can be recommended for industrial applications in the more proper choice of the tool electrode material for electrical discharge machining design in the context of structural requirements for the final product working surfaces to improve their functionality in the actual conditions of exploitation.

## Figures and Tables

**Figure 1 materials-14-03189-f001:**
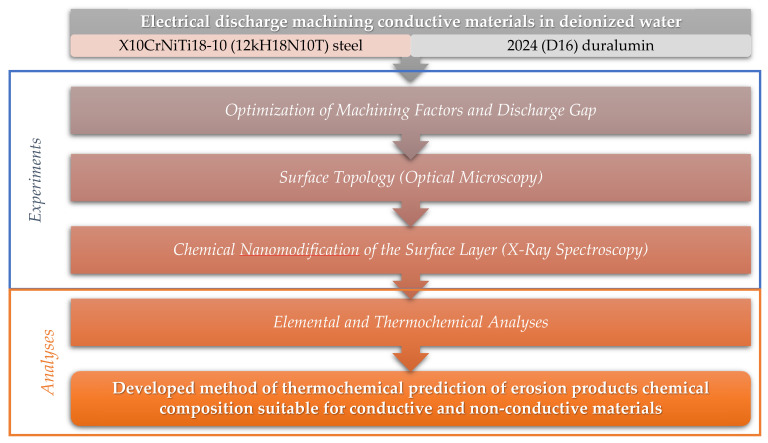
Flow chart of the stepwise procedure of experiments and analyses carried out.

**Figure 2 materials-14-03189-f002:**
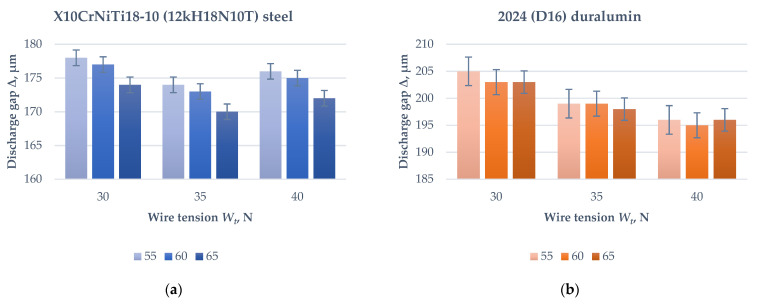
Histograms of the measured discharge gaps on electrical discharge machining factors *V_o_* and *W_t_*: (**a**) X10CrNiTi18-10 (12kH18N10T) steel; (**b**) D16 (AA2024) allo2024 (D16) duralumin.

**Figure 3 materials-14-03189-f003:**
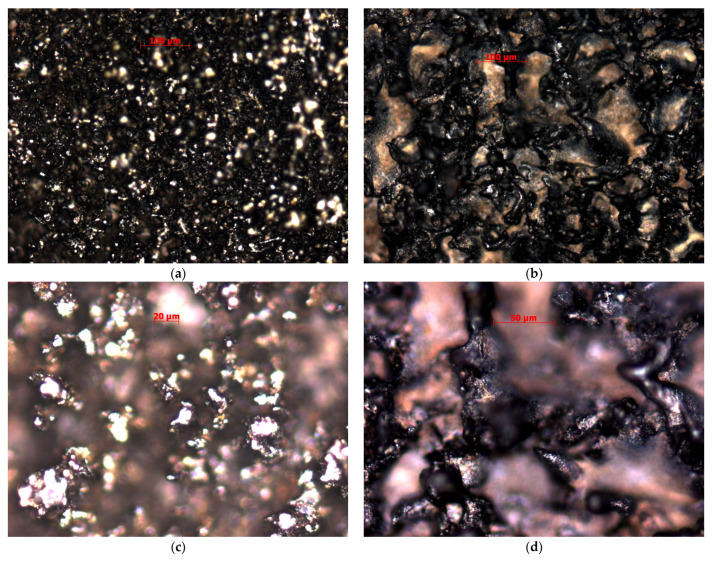
Topology the samples after electrical discharge machining with a brass wire tool electrode in water: (**a**) X10CrNiTi18-10 (12kH18N10T) steel, 20×; (**b**) 2024 (D16) duralumin, 20×; (**c**) X10CrNiTi18-10 (12kH18N10T) steel, 50×; (**d**) 2024 (D16) duralumin, 50×.

**Figure 4 materials-14-03189-f004:**
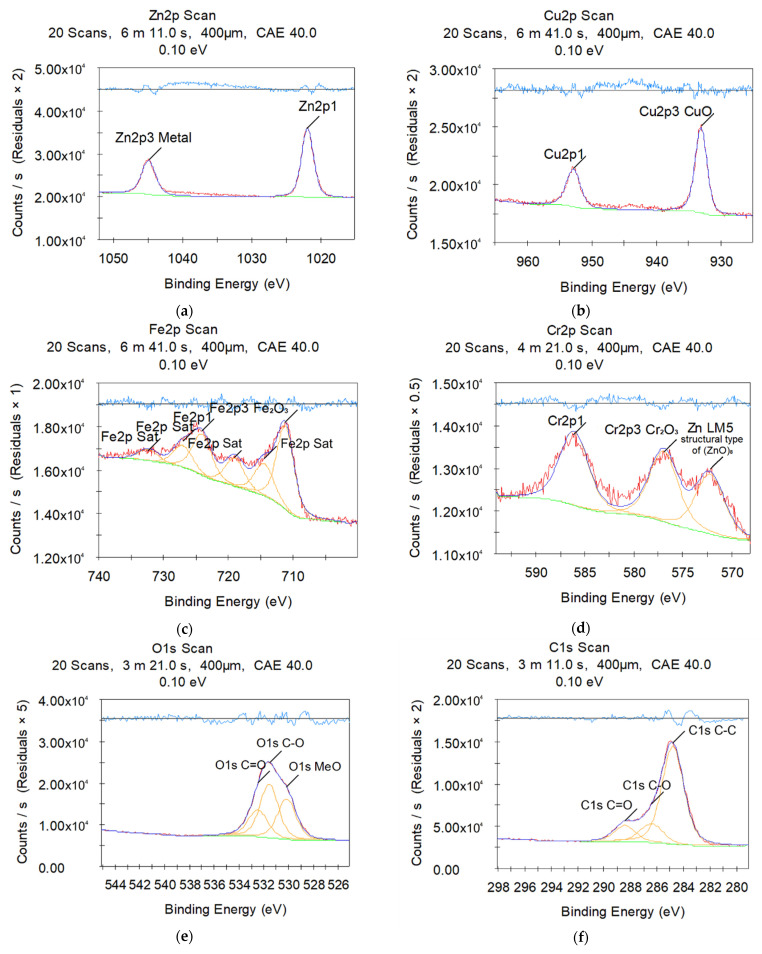
X-ray photoelectron spectroscopy of machined surfaces of X10CrNiTi18-10 (12kH18N10T) steel workpiece in water medium: (**a**) zinc; (**b**) copper; (**c**) iron; (**d**) chromium; (**e**) oxygen; (**f**) carbon.

**Figure 5 materials-14-03189-f005:**
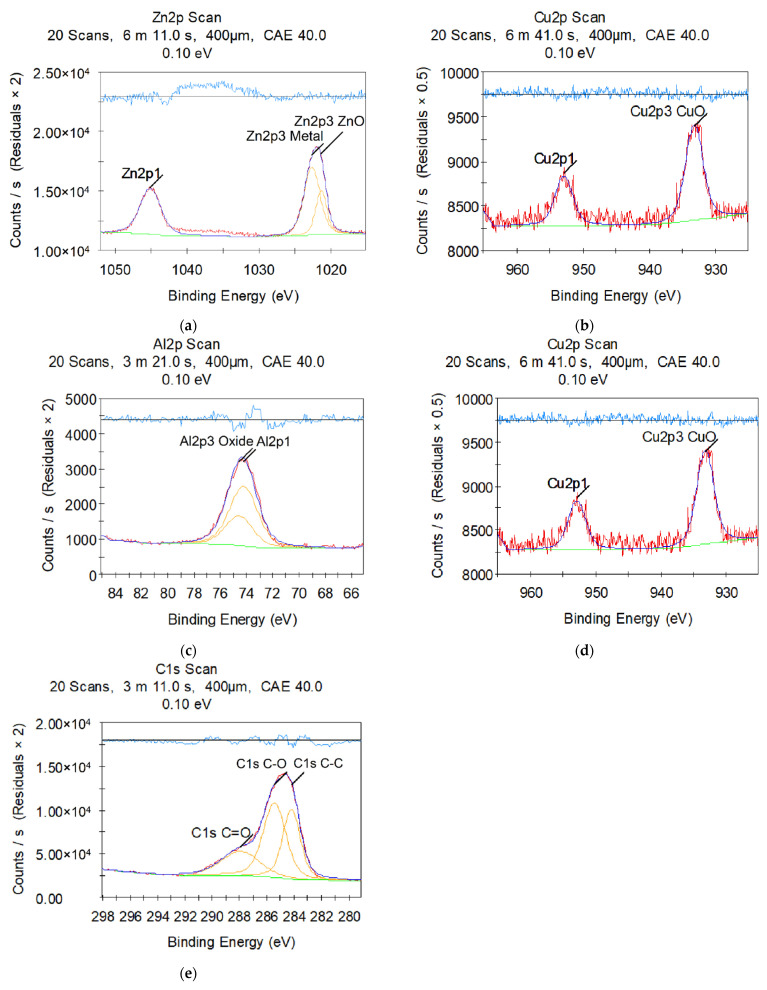
X-ray photoelectron spectroscopy of machined surfaces of 2024 (D16) duralumin workpiece in water medium: (**a**) zinc; (**b**) copper; (**c**) aluminum; (**d**) oxygen; (**e**) carbon.

**Figure 6 materials-14-03189-f006:**
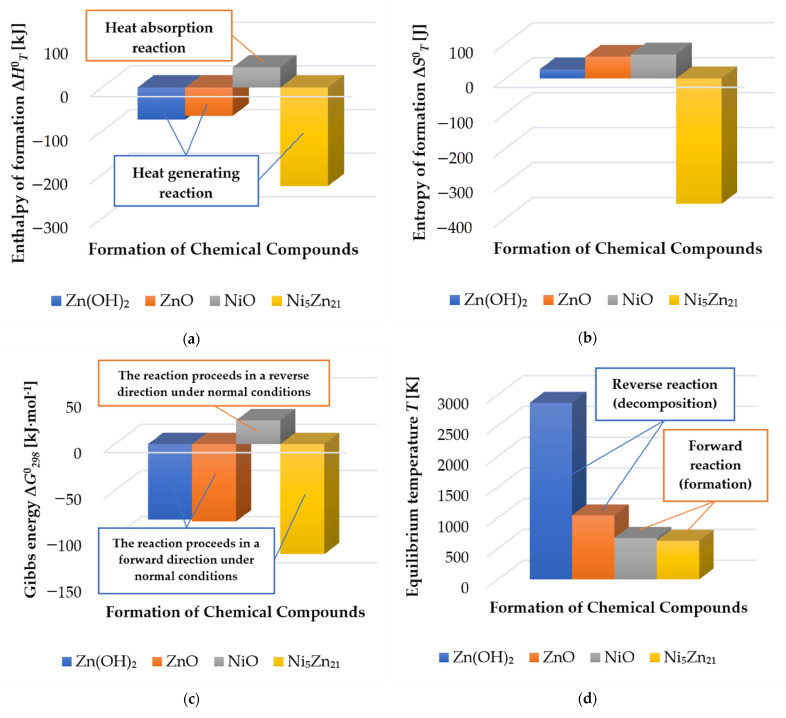
Thermochemical parameters of chemical reactions of nickel and zinc in a water medium: (**a**) enthalpy of formation; (**b**) entropy of formation; (**c**) Gibbs energy; (**d**) equilibrium temperatures.

**Table 1 materials-14-03189-t001:** Main characteristics of a Seibu M500S wire electrical discharge machine.

Characteristic, Measuring Unit	Value
Max motions in working area X × Y × Z, mm	500 × 350 × 310
Max conical machining angle, degrees	±10
Max weight of workpiece, kg	800
Axes positioning accuracy, µm	±1–2
Achievable roughness parameter *R_a_*, µm	0.4
Dielectric medium	Deionized water
Machine body	Solid, grey cast iron
Max power consumption, kW	not confined

**Table 2 materials-14-03189-t002:** Electrical discharge machining factors (Seibu M500S machine).

Factors	Values ^1^
Operational voltage (*V_o_*)	55, 60, 65
Auxiliary voltage (*V_g_*)	32
Operational current strength (*I*)	8
Auxiliary current (reverse circuit) (*I_a_*)	43
Discharge time to time off ratio (*T_off_*)	6
Time intermittent pause (*T_i_*)	305
Feed rate (*R_f_*)	Adaptive
Wire tool rewinding speed (*R_r_*)	35
Wire tool tension (*W_t_*)	30, 35, 40
Dielectric pressure in nozzles (*F_d_*)	245

^1^ Provided in equivalent units of the machine.

**Table 3 materials-14-03189-t003:** Chemical content of X10CrNiTi18-10 (12kH18N10T) steel.

Element	Fe	Cr	Ni	Ti	Si	S	Mn	Cu	P	C
wt.%	Balance	17–19	9–11	~0.8	<0.8	<0.02	<2.0	<0.03	<0.035	~0.12

**Table 4 materials-14-03189-t004:** Chemical content of 2024 (D16) duralumin.

Element	Al	Cu	Mg	Mn	Fe	Si	Zn	Ni	Ti
wt.%	90.8–94.7	3.8–4.9	1.2–1.8	0.3–0.9	<0.5	<0.5	<0.3	<0.1	<0.1

**Table 5 materials-14-03189-t005:** Electrical and thermal properties of some materials at +20 °C.

Materials	Specific Electrical Resistance $\rho \;\left[\frac{\Omega \cdot {\mathrm{mm}}^{2}}{m}\right]$	Melting Point *T_m_* [°C]	Decomposition Point *T_d_* [°C]
X10CrNiTi18-10 (12kH18N10T) steel	0.725	1420–1800	-
2024 (D16) duralumin	0.052	650	-
CuZn35 brass alloy (annealed)	0.065	920	-
Nickel ^1^	0.087	1453–1455	2732–2913 (boiling)
Zinc ^1^	0.059	419.6	906.2 (boiling)
NiO ^1^	-	1682–1955	1230 (sublimation)
ZnO ^1^	~0.01 × 10^14^	1975 (decomposition)	1800 (sublimation)

^1^ Given for references.

**Table 6 materials-14-03189-t006:** X-ray peaks of the X10CrNiTi18-10 (12kH18N10T) steel sample machined surfaces.

Chemical Element	Binding Energy *E*	Peak of Binding Energy, eV	Atomic %
Oxygen	O1s	532.5	29.31
Zinc	Zn2p3	1022.7	3.44
Carbon	C1s	286.2	58.07
Copper	Cu2p3	934.3	1.4
Iron	Fe2p	712.3	1.68
Nitrogen	N1s	400.8	2.21
Magnesium	Mg1s	1305.5	0.22
Sodium	Na1s	1072.4	0.51
Chlorine	Cl2p	200.8	0.89
Chromium	Cr2p	575.2	2.25

**Table 7 materials-14-03189-t007:** X-ray peaks of the 2024 (D16) duralumin sample machined surfaces.

Chemical Element	Binding Energy *E*	Peak of Binding Energy, eV	Atomic %
Oxygen	O1s	531.9	36.66
Zinc	Zn2p3	1022.2	1.87
Magnesium	Mg1s	1304.3	4.38
Carbon	C1s	285	40.37
Aluminum	Al2p	74.3	10.42
Nitrogen	N1s	400	2.59
Calcium	Ca2p	352	2.12
Copper	Cu2p3	933.4	0.25
Chlorine	Cl2p3	198.9	1.35

**Table 8 materials-14-03189-t008:** Calculated enthalpy of formation of some Zn- and Ni-containing substances.

Chemical Reaction	Chemical Composition of Erosion Product	Amount of Substance *n* [mol]	Enthalpy of Substances Δ*H*^0^_298_ [kJ·mol^−1^]	Enthalpy of Formation Δ*H*^0^*_T_* [kJ]
(2)	Zn(OH)_2_	$1+2\overset{T}{\to }1+1$	$0-285.8\overset{T}{\to }-645.4+0$	[1·(−645.4)+1·0]−[1·0+2·(−285.8)]=−73.8
(3)	ZnO	$1+1\overset{T}{\to }1+1$	$0-285.8\overset{T}{\to }-350.8+0$	[1·(−350.8)+1·0]−[1·0+1·(−285.8)]=−65.0
(4)	NiO	$1+1\overset{T}{\to }1+1$	$0-285.8\overset{T}{\to }-239.7+0$	[1·(−239.7)+1·0]−[1·0+1·(−285.8)]=46.1

**Table 9 materials-14-03189-t009:** Calculated entropy of some Zn- and Ni-containing substances and equilibrium temperatures.

Chemical Reaction	Amount of Substance *n* [mol]	Entropy of Substances Δ*S*^0^_298_ [J·mol^−1^]	Entropy of Formation Δ*S*^0^*_T_* [J]	Gibbs Energy Δ*G*^0^*_298_* [kJ·mol^−1^]	Equilibrium Temperature *T* [K]
(2)	$1+2\overset{T}{\to }1+1$	$41.6+70.1\overset{T}{\to }77.0+130.52$	[1·77.0+1·130.52]−[1·41.6+2·70.1]=25.72	−73.8−298·0.02572=−81.46	$\frac{\left|73.8\right|}{0.02572}=2869.36$
(3)	$1+1\overset{T}{\to }1+1$	$41.6+70.1\overset{T}{\to }43.6+130.52$	[1·43.6+1·130.52]−[1·41.6+1·70.1]=62.42	−65.0−298·0.06242=−83.60	$\frac{\left|65.0\right|}{0.06242}=1041.33$
(4)	$1+1\overset{T}{\to }1+1$	$29.9+70.1\overset{T}{\to }38.0+130.52$	[1·38.0+1·130.52]−[1·29.9+1·70.1]=68.52	46.1−298·0.06852=25.68	$\frac{\left|46.1\right|}{0.06852}=672.80$

## Data Availability

Data available in a publicly accessible repository.

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
