# Peer review of "Elemental and Thermochemical Analyses of Materials after Electrical Discharge Machining in Water: Focus on Ni and Zn"

_materials, 2021, doi:10.3390/ma14123189_

Round 1

Reviewer 1 Report

The publication is written in the correct language. It is a typical application work of a technical nature. Please enhance the scientific descriptions. I don't see any need to improve this work. If the editor finds that the technology work is suitable for the journal, then this publication is appropriate. 

Author Response

Dear Reviewer,
Thank you very much for your kind evaluation of our article. If you point where we should enhance the scientific descriptions, we will do it with great pleasure.
We wish you much success in your current projects.

Kind regards,
Authors.

Reviewer 2 Report

Journal: Material

Ref manuscript ID materials-1230954

Manuscript title “Elemental and Thermochemical Analyses of Materials after 2

Electrical Discharge Machining in Water: Focus on Ni and Zn’’

Comments to authors:

This paper reads very comfortable. According to my opinion, this paper is not suitable for publication in the journal in its current form only after appropriate major revisions are made.
The comments are in the following:

  • Add the list of symbols as nomenclature before the introduction section? Since the document contains several incomprehensible symbols and characters and all symbols used in the text , tables and equations.
  • The manuscript is in need of substantial editing for grammatical errors. Could you please check the grammatical errors in the manuscript in several places. Use of articles 'the', 'a' and 'an' needs to be corrected at some places
  • In abstract the author must provide the highlight of this study.
  • the authors should explain the difference between this document and that published by the same authors in the journal J. Manuf. Mater. Process which is cited as [37] in the reference section
  • the sequential order of the equations is not respected in the document, I see that the authors have inserted directly in page 5, the equation (4), where are the other equations (1-2-3)
  • Electrochemical Discharge Machining (ECDM) uses the principle of thermal fusion and chemical dissolution for machining materials. a light description of this process is strongly recommended in the introduction section
  • On page 1, line 32-33, The first sentence of the introduction (Electrical discharge machining is a well-known technology used for machining conductive materials [1-4].) is copied directly from the article published by the same authors number [37] in reference, but the references [1-4] are different, why ???, please check these references or change this sentence
  • The authors presented the results, but they didn't give a sufficient discussion for the results. This makes the paper look like a lab report rather than a research paper.
  • There are many research papers study the same problem which investigated in the present paper. What is exactly the new point of this work?
    The authors should focus to clarify this issue in the paper.
  • Authors should explain more about the novelty of their work in introduction,
  • The authors stated in the introduction section the following “Indeed, the nature of erosional destruction is of a mixed nature, but not thermo-mechanical, it is thermochemical.” It is therefore requested to explain this statement in depth and why ?
  • From the introduction, it seems that the authors know little about what has been done by others in this field. A detailed literature review is strongly recommended.
  • A  simple flow chart is required to represent the steps-wise procedure followed for carrying out the all analysis, test performance , test produce and test results.
  • Key assumptions and their implications could have been elaborated
  • Details of the test procedure should be described in detail .The authors have to explain what method was used and why.
  • The authors should think over the real significance of their results and try to rewrite this section to improve understanding of the conclusions
  • The quality of the figures in this document needs to be improved; the figures need to be larger in size so the data and labels can be clearly read.
  • The introduction section is adequate but needs some language revision.
  • The introduction should provide a clear statement of the problem, the relevant literature on the subject, and the proposed approach or solution. It is be understandable to colleagues from a broad range of scientific disciplines. the following article must be added to the author's contribution:

-Thermomechanical behavior of dry contacts in disc brake rotor with a grey cast iron composition,Thermal Science 17(2) ,2013, pp.599-609

  • In 'Result and Discussion' authors have noted observations. But it is suggested that to provide physical explanations of all obtained results which can enrich the quality of the paper.
  • The authors must provide a greater discussion of the results
  • The conclusion should be clear.

The conclusions should be re-written, they should be specific and should not include as the initial paragraphs, which seems a summary, it must be enriched about discussion on solving problem

Altogether, the paper needs modification to be suitable for the standards required for publication; therefore I recommend that it required to Major revision”.

Author Response

Response to Reviewer 2 Comments

Dear reviewer,

Thank you very much for your kind evaluation of our work. We do agree with all your proposals and comments and have modified the manuscript according to them. The revised fragments are marked yellow.

We hope that with your comments the manuscript will be suitable for publishing in Materials and will attract many potential readers of the journal. 

Kind regards,

Authors.

Point 1: Add the list of symbols as nomenclature before the introduction section? Since the document contains several incomprehensible symbols and characters and all symbols used in the text , tables and equations.

Response 1: Thank you for your proposal. We have added the nomenclature of used symbols after the introduction. Technically, it was not possible to add it before the introduction.

Point 2: The manuscript is in need of substantial editing for grammatical errors. Could you please check the grammatical errors in the manuscript in several places. Use of articles 'the', 'a' and 'an' needs to be corrected at some places.

Response 2: Thank you for pointing it; we have revised the manuscript. The revised places are marked yellow. The grammar check shows the overall score of 99 of 100 percent of performance. The report is attached.

Point 3: In abstract the author must provide the highlight of this study.

Response 3: Thank you, the abstract is revised.

Point 4: the authors should explain the difference between this document and that published by the same authors in the journal J. Manuf. Mater. Process which is cited as [37] in the reference section

Response 4: Thank you for noticing it. These are two different papers:

1) The main difference is in used materials. The mentioned article was about ceramic nanocomposites (Al2O3+30%TiC nanoparticle), including the stage of sintering ceramic. This article is about conductive materials in general and metals in particular (anti-corrosion steel of austenite class and duralumin).

2) The second difference is in comparison of the dielectric medium (water or oil) in the formation of nanostructure of nanocomposites, focus on avoiding aluminum carbide in the first article, and comparison of two structural material’ components interactions with the material of wire tool electrode (brass) to provoke the formation of superconductive intermetallic erosion dust.

3) The difference is also in using measures of vibroacoustic diagnostics and EDX mapping in the first article, and unique and proposed for the first time thermochemical approach with calculation all possible thermochemical parameters and equilibrium temperature in this article. The theoretical explanation of the method is announced to be published in Materials and Design.

4) The similarities are in optical and scanning electron microscopy, X-ray spectroscopy, and calculation of discharge gap.

The mentioned article has open access, and it can be easily checked by anyone if there any similar workpiece material, the structure of the manuscript, and its content, including pictures, graphs, and approaches.

Point 4: the sequential order of the equations is not respected in the document, I see that the authors have inserted directly in page 5, the equation (4), where are the other equations (1-2-3)

Response 4: Thank you, it is revised.

Point 5: Electrochemical Discharge Machining (ECDM) uses the principle of thermal fusion and chemical dissolution for machining materials. a light description of this process is strongly recommended in the introduction section

Response 5: Thank you, it is revised.

Point 6: On page 1, line 32-33, The first sentence of the introduction (Electrical discharge machining is a well-known technology used for machining conductive materials [1-4].) is copied directly from the article published by the same authors number [37] in reference, but the references [1-4] are different, why ???, please check these references or change this sentence

Response 6: Thank you, there is a different research subject, the authors would like to focus on other things. Further, some of these references were used in the manuscript. The sentence is revised.

Point 7: The authors presented the results, but they didn't give a sufficient discussion for the results. This makes the paper look like a lab report rather than a research paper.

Response 7: Thank you, we have improved the discussion. Finally, we have decided to revise the manuscript completely to make the main subject clearer and remove all unnecessary data. Now there are 17 pages only the data related to the research subject.

We found it was already quite an extensive discussion since there were 25 pages of the manuscript, and 4 of them were devoted to the very complex discussion that goes very deep in thermochemistry when some scientists publish the manuscript of only nine pages without discussion (https://www.mdpi.com/1996-1944/14/11/2770 ). There are 20 pages of the manuscript and two pages of discussion in the mentioned J. Manuf. Mater. Process cited as [37]. We would also like to add that we did not want to overload the manuscript with unnecessary data since this is a regular manuscript. 

Point 8: There are many research papers study the same problem which investigated in the present paper. What is exactly the new point of this work?

The authors should focus to clarify this issue in the paper.

Response 8: Thank you for pointing it. Up to now, there is no single work devoted to the analysis of thermochemical parameters between electrodes in the presence of heat, especially reactions with the formation of intermetallic bonds. For the first time, the electrical but the chemical nature of the bonds was mentioned by the Lazarenko spouses themselves, who developed the method of electrical discharge machining in 1942. Since then, only A. Kalka and Y. Guo were mentioned in their articles on chemical transformations. If A. Kalka plunged into this issue in sufficient detail from the material science point of view (eutectic, crystal formation), but only touched on the topic of EDM, then Y. Guo casually mentions the thermochemical nature in the context of the processing of insulating ceramics. None of the authors who investigate electrical erosion have gone so far as to explain the resulting modifications of the workpiece layers from a thermochemical point of view. There is no explanation why the assisting alumina powder does not allow intensifying the machining of steel, why sparks are formed during the processing of stainless steel and a cloud of black sediment, why insulating zirconium dioxide can be machined in the hydrocarbon medium using the assistant electrode technique. At the same time, aluminum oxide and nitride are very laborious to process, etc. This article presents a new, unique approach that was never previously presented to understanding the profound nature of electrical erosion after the discharge channel has formed. It makes it possible to evaluate the erosion products and select them based on the properties of the electrical conductivity of the erosion products, to calculate theoretically the probability of the formation of certain erosion products based on the calculated Gibbs energy, entropy, and enthalpy of formation, equilibrium temperature.

Point 9: Authors should explain more about the novelty of their work in introduction,

Response 9: The scientific novelty of the work is in:

  • The new data on thermochemical and thermophysical phenomena that occurred between electrodes and water medium in the presence of plasma heat,
  • Completed thermochemical analyses of chemical interactions of the surface and near-surface layers after electrical discharge machining.

This paragraph is revised.

Point 10: The authors stated in the introduction section the following “Indeed, the nature of erosional destruction is of a mixed nature, but not thermo-mechanical, it is thermochemical.” It is therefore requested to explain this statement in depth and why ?

Response 10: Thank you for your attention to the details. Since electrical discharge machining, electrochemical machining, and electrochemical discharge machining are contactless machining methods, there is always an interelectrode gap between the tool and workpiece electrodes and the physical contact is absent (https://en.wikipedia.org/wiki/Electrochemical_machininghttps://en.wikipedia.org/wiki/Electrical_discharge_machininghttps://journals.sagepub.com/doi/abs/10.1177/0954405418798865?journalCode=pibb#:~:text=The%20electrochemical%20discharge%20machining%20(ECDM,electrical%20discharge%20machining%20(EDM). ). It allows machining materials despite their hardness or ductility with high precision. Thus, it cannot have any thermomechanical nature since there is no mechanics (https://en.wikipedia.org/wiki/Mechanics ), when thermochemical nature is the novelty of this research proved by X-ray spectroscopy and thermochemical equations of a thermally isolated system (Gibbs energy, entropy, and enthalpy of formation, equilibrium temperature) (https://en.wikipedia.org/wiki/Thermochemistryhttps://en.wikipedia.org/wiki/Thermally_isolated_system ).

Point 11: From the introduction, it seems that the authors know little about what has been done by others in this field. A detailed literature review is strongly recommended.

Response 11: Thank you for your efforts in making the manuscript better. We added a few more references. We would like to note that there is plenty of reviews concerning electrical discharge machining but a few works devoted to thermochemical nature, which is part of our novelty and our research focus. Just electrical discharge machining of structural materials was never a problem, and achievements in roughness and discharge gap in this context depend more on the particular machine than on thermochemistry.

Point 12: A  simple flow chart is required to represent the steps-wise procedure followed for carrying out the all analysis, test performance , test produce and test results.

Response 12: Thank you for your suggestion. A flow chart of the experiments' stepwise procedure and analyses is added (Figure 1).

Point 13: Key assumptions and their implications could have been elaborated.

Response 13: Thank you. The practical significance of the work is added.

Point 13: Details of the test procedure should be described in detail .The authors have to explain what method was used and why.

Response 13: Thank you. We have improved the section of methods.

Point 14: The authors should think over the real significance of their results and try to rewrite this section to improve understanding of the conclusions

Response 14: Thank you. The conclusions are revised.

Point 15: The quality of the figures in this document needs to be improved; the figures need to be larger in size so the data and labels can be clearly read.

Response 15: Thank you. The quality of the figures is improved.

Point 16: The introduction section is adequate but needs some language revision. The introduction should provide a clear statement of the problem, the relevant literature on the subject, and the proposed approach or solution. It is be understandable to colleagues from a broad range of scientific disciplines. the following article must be added to the author's contribution:

-Thermomechanical behavior of dry contacts in disc brake rotor with a grey cast iron composition,Thermal Science 17(2) ,2013, pp.599-609

Response 16: Thank you. The introduction is revised. The recommended reference was not added due to possible conflict of interests (ethical issue since we do not know if the author of this publication is our reviewer or not). Moreover, the most critical Thermomechanical behavior of dry contacts in disc brake rotor with a grey cast iron composition is not related to the research subject of the current manuscript. Our manuscript is not about grey cast iron, not about dry contact, and not about thermomechanical behavior.

Point 17: In 'Result and Discussion' authors have noted observations. But it is suggested that to provide physical explanations of all obtained results which can enrich the quality of the paper.

Response 17: Thank you. The thermophysical nature is explained in the discussion.

Point 18: The authors must provide a greater discussion of the results

Response 18: Thank you. The discussion is revised.

Point 19: The conclusion should be clear. The conclusions should be re-written, they should be specific and should not include as the initial paragraphs, which seems a summary, it must be enriched about discussion on solving problem

Response 19: Thank you. The conclusions are revised.

Reviewer 3 Report

In this manuscript, the authors investigated material removal mechanism, physical and chemical nano transformations that occurred on surface and subsurface layers during electrical discharge machining on steel and duralumin by brass tool. Overall, this reviewer found the manuscript very difficult to read which could be due to language barrier. As such, some discussion information may have been lost in translation. The authors however provide a detailed materials and methods section. The content seems reasonable for publication but it has to go through revision and the authors should consider involving someone who to help with the language so ensure that the manuscript is well written.

Below are specific issues/concerns that this particular reviewer has:

  1. This statement “The thermochemical nature of electrical erosion was firstly proposed by the authors of the method, Prof. B. Lazarenko and Prof. N. Lazarenko (1942), and was repeated again only by A. Calka in Nature (2002) and Y. Guo in Materials and Manufacturing Processes (2014)” should be removed from the abstract, it belongs in the introduction/background section.

  1. There is no good elaboration and discussion on the experimental results, case in point, the authors state that “The measured discharge gap is 45 – 60 μm for X10CrNiTi18-10 (12kH18N10T) steel, and 105 – 120 μm for 2024 (D16) duralumin”  without discussing why there is a difference in the results obtained on the two samples.

  1. Figures and figure captions do not explain the details that authors wish to express (e.g figure 2). In Figure 5, the distinction between the mechanical rupture location and melting traces of the short circuit (in Figure 5 a & b) is not clear to this reviewer. Is figure 12 a schematic illustration or a figure from literature? Is the thickness of the “secondary structure” known?

  1. Adding results from mechanical properties characterization (hardness and modulus, wear test to quantify war resistance etc.) of the subsurface layer of the tool and samples would aid in advancing the story on nanomodification of the surface sample layer by using different components of the machining processes. The reviewer recommends the authors to consider adding these results.

  1. Section 6, this reviewer wonders how relevant it is to the current manuscript if the patents are not cited in the manuscript. The section does not add any value to the reader therefore should be deleted entirely.

Author Response

Response to Reviewer 3 Comments

Dear reviewer,

Thank you very much for your kind evaluation of our work. We do agree with all your proposals and comments and have modified the manuscript according to them. The revised fragments are marked green.

We hope that with your comments the manuscript will be suitable for publishing in Materials and will attract many potential readers of the journal.

Kind regards,

Authors.

Point 1: This statement “The thermochemical nature of electrical erosion was firstly proposed by the authors of the method, Prof. B. Lazarenko and Prof. N. Lazarenko (1942), and was repeated again only by A. Calka in Nature (2002) and Y. Guo in Materials and Manufacturing Processes (2014)” should be removed from the abstract, it belongs in the introduction/background section.

Response 1: Thank you for your correct proposal. The abstract is revised.

Point 2: There is no good elaboration and discussion on the experimental results, case in point, the authors state that “The measured discharge gap is 45 – 60 μm for X10CrNiTi18-10 (12kH18N10T) steel, and 105 – 120 μm for 2024 (D16) duralumin”  without discussing why there is a difference in the results obtained on the two samples.

Response 2: Thank you for pointing it. The difference is related to the electrical properties of these materials. As can be seen from table 5 they have different in ~14 times values of specific electrical resistance (0.725  for steel and 0.052  for duralumin) that explains the difference in the discharge gap. Since duralumin is more conductive (electrical conductivity is the reciprocal of electrical resistance), the breakdown of the dielectric medium (deionized water) occurs at a greater distance than for stainless steel. According to generally accepted information about the typically recommended spark gaps for materials and electrical resistivity values, where the lowest value of the conductive materials refers to graphite, which also exhibits electrical anisotropy, and the highest value to silver, metal carbides, superconducting intermetallics. The nickel present in stainless steel improves the electrical conditions in the interelectrode gap, but not enough to be comparable to the more electrically conductive material - duralumin. Further research is planned to be devoted to the elemental and thermochemical study of two materials with similar winter composition and different nickel content when treated with a brass electrode in water or hydrocarbons.

Point 3: Figures and figure captions do not explain the details that authors wish to express (e.g figure 2). In Figure 5, the distinction between the mechanical rupture location and melting traces of the short circuit (in Figure 5 a & b) is not clear to this reviewer. Is figure 12 a schematic illustration or a figure from literature? Is the thickness of the “secondary structure” known?

Response 3: Thank you, we have removed all unclear figures from the manuscript. Figure 12 was our creation based on obtained data, experience, and knowledge (there is no data on it in the literature). Since it is a scientific article, all our pictures are original and were drawn only by our research team. Figure 5 will be explained in our subsequent work since the article was already extensive, and we decided to shorten it. Figure 2 was devoted to illustrating the formed slot (kerf) by a single pass of the wire tool that can be interesting for the specialists in the area of the research.

Point 4: Adding results from mechanical properties characterization (hardness and modulus, wear test to quantify war resistance etc.) of the subsurface layer of the tool and samples would aid in advancing the story on nanomodification of the surface sample layer by using different components of the machining processes. The reviewer recommends the authors to consider adding these results.

Response 4: Thank you for your kind suggestion. It can be a good proposal for our next research that requires a little bit more time and attention. We appreciate very high your competence in this area and your proposal. Indeed, this topic deserves more detailed research.

Point 5: Section 6, this reviewer wonders how relevant it is to the current manuscript if the patents are not cited in the manuscript. The section does not add any value to the reader therefore should be deleted entirely.

Response 5: Thank you, we have used in our experiment some techniques developed and patented previously. The section is removed.

Round 2

Reviewer 2 Report

Journal:  Materials-MDPI

Ref manuscript ID  materials-1230954

Manuscript title “Elemental and Thermochemical Analyses of Materials after 2

Electrical Discharge Machining in Water: Focus on Ni and Zn

Dear 
Ms. Divena Dai

Assistant Editor

Materials-MDPI

Comments:The authors have satisfactorily responded to all my questions and made the necessary changes to the manuscript. The revised version of the manuscript appears to be good.

My final recommendation: Accept With No Changes

Thank you again for giving me a chance and inviting me to review this document.

Best regards

Author Response

Dear Reviewer,
Thank you very much for your kind evaluation of our article, help, and assistance in making our manuscript better.
We wish you much success in all your projects and work.

Kind regards,
Authors.

Reviewer 3 Report

Overall, the authors have tried their best to improve the manuscript. However, there are still minor editing require and grammatical errors that should be considered before publication. Here are a few suggestions:

  1. Nomenclature table need to be directly after the abstract. This way, any nomenclature is explained before it is used in the main text.
  2. The paragraphs on page 3 (lines 136 -147) could be summarized into single concise and clear paragraph
  3. Figure 2 is not mentioned anywhere in the text.
  4. Figures should fit on a single page (no separation like figures 4, 5 and 6). Also, tables and table captions should be on the same page (see Table 7).

Author Response

Response to Reviewer 3 Comments

Dear reviewer,

Thank you very much for your kind evaluation of our work. We do agree with all your proposals and comments and have modified the manuscript according to them. The revised fragments are marked blue.

We hope that with your comments the manuscript will be suitable for publishing in Materials and will attract many potential readers of the journal.

Kind regards,

Authors.

Point 1: Nomenclature table need to be directly after the abstract. This way, any nomenclature is explained before it is used in the main text.

Response 1: Thank you for your kind suggestion. The table was placed before the introduction. It should be noted that it is pretty tough to do in the current template. We tried to split the table into two pieces, and we hope that editors will help us place it most suitably and beautifully for a better presentation.

Point 2: The paragraphs on page 3 (lines 136 -147) could be summarized into single concise and clear paragraph

Response 2: Thank you, it is revised.

Point 3: Figure 2 is not mentioned anywhere in the text.

Response 3: Thank you for noticing it. It is revised.

Point 4: Figures should fit on a single page (no separation like figures 4, 5 and 6). Also, tables and table captions should be on the same page (see Table 7).

Response 4: Thank you, it is revised.

This manuscript is a resubmission of an earlier submission. The following is a list of the peer review reports and author responses from that submission.

Round 1

Reviewer 1 Report

Type of the Paper (Article, Review, Communication, etc.)

Tool Electrode Behavior under Pulses, Wear and Surface Nanomodification atElectrical Discharge Machining

Sergey N. Grigoriev, Marina A. Volosova, Anna A. Okunkova1, Sergey V. Fedorov KhaledHamdy, Pavel A. Podrabinnik, Petr M. Pivkin, Mikhail P. Kozochkin and ArturN.Porvatov

The title of the publication is appropriate.
The abstract is well built. The information contained in it presents exactly what problems will be undertaken at work. I would like to mention that the abstract shows that the work is of a technical nature with scientific elements. The results of these studies can form the basis of further applications.
The theoretical introduction accurately presents the problem undertaken at work. The applied literature is directly related to the subject of the publication.
The methodology and approach to the topic is good.
The applied approach of studying the wear process of the tool electrode during machining is interesting. Especially because of the possibility of precise monitoring taking into account the vibrations. Such a study can be used in the future when designing modern devices. As a result of the research, it was found that the surface of the material may change due to the cooling medium used. This observation is accurate and would require careful analysis. However, at this stage it should be stated that with the mass commercial use of this factor, it is rather impossible to eliminate. The coolants used are standard and I don't see the need to look for others. The resulting layer may or may not be desired.
Summing up, the work is written correctly. Let me mention that it is not a strictly scientific study, but rather a technical one. Due to the fact that I prefer such works, I recommend it for publication. As a remark, I can only ask the authors whether they have carefully studied the influence of the nano-layer.

Reviewer 2 Report

The manuscript gave some observations concerning the process of the electrcal discharge machining (EDM) on variuous conventional materials. However, these results could be obtained naturally when we used the processing (EDM) on the materials (ceramics, metals, and composites). Authors have to give the explanation of correlation between specific information of the materials and electrical discharge machining. Authors have to give the specific information of the materials used in this study by data of analysis themselves. I cannot find somthing new key purpose in this study. The manuscript in its present state should not be published in Materials. However, some results in this study is useful to engineers who are interested in EDM processing.

  • What is the scientific key purpose of the study?
  • What is nanocomposites?
  • The comments, "The current state of the mentioned research subject.......present. That can be related with the low....electrodes," is Author's' opinion or not? Some reference is needed. 
  • This manuscript requires further editing to conform it to correct scientific English.

Reviewer 4 Report

Although the results seem interesting, the paper lacks consistency and is very hard to read. The authors should probably request the advice of a native English speaker to rephrase and reformulate the paper. In its present form, I cannot recommend this paper for publication.

Reviewer 5 Report

This paper refers to an investigation into the process of EDM cutting of two materials (stainless steel and aluminum alloy). The effect of different machines and process parameters on the cutting-wire damage and EDM process stability was investigated. 

In my opinion, the paper should be rejected due to material engineering-drawbacks and resubmitted to the Processes journal (or Sensors). Generally, the main plot of the manuscript describes the "EDM process and process parameters". The main plot of the paper contains EDM process characterization and focused on the process stability. The quality of the "tool" and "workpiece" are only the indicators of the "EDM process". Thus, the Materials is not an appropriate journal for that paper mainly due to limited materials investigation.

Generally, the paper is too long and some of the paragraphs could be deleted. The scientific goal of the paper should be much more emphasized. It is difficult to catch the idea of the paper. The content of the paper must be rearranged according to the Processes of Sensors journals requirements. 

Also from the material-point of view I have some comments on the paper: 

  1. tab1 - the max power consumption and machine body should be given for all tested machines.
  2. Is it scientific accurate to compare the results obtained from testing in 3 different machines (3 different test cutting conditions)? I think that this is main methodological-error that affects the comparison of the results. These machines have different cutting-accuracy, different cutting envirnoment etc and adjustable (unknown) selected parameters - I think that this should be toughly justified in the manuscript. 
  3. The meaning of that phrase is not clear - should be justified "The factors were changed manually and gradually during processing until the moment of
    162 wire breakage to obtain a sample of wire electrode for microscopic research". 
  4. Table 3 - the carbon content in the steel can not equal 12% - it must be improved.
  5. What is the source of the drawings given in fig3? How it corresponds to the aim of the work?
  6. The figure 11 - is not clear. The damaged surface layer is not clear.
  7. In fig12 - authors write "surface films of metastable solid solution or eutectics" which must be stated by the XRD and metallographic investigations. The XRD spectra of as-received materials should be compared to the XRD spectra done in the cutting area.
  8. In fig12 - I do not agree with the phrase "thermal cracks"; why you claim that cracks are a thermal effect? Maybe it is a mechanical effect. 
  9. Section 3.4 - tool characterization - the aim of that section is not clear. Also, the fractographic description must be improved according to the failure-analysis principles.
  10. Table 5 - the measurements should be done for more than one sample - then the scientific accuracy will be obtained. The failure of the tool electrode is random - therefore the statistics should be employed. 
  11. Fig 15 - should be deleted - do not provide any important data. Simply you confirm the chemical composition of cutting wire. What for? 
  12. Fig 16 - presents the surfaces after cutting but only the one parameters "medium: oil / water" is given. Now, it does not provide any scientific information. I think that process parameters (which are not stated precisely) effects on the surface topography. All tested surfaces with different feeds, voltages, and other input parameters should be presented.
  13. Fig 17 - is difficult to read and sorry for that but I cannot understand its meaning. Please improve it. 
  14. This phrase should be justified by tough materials research: "SEM-microphotographs (Figure 12) excludes any
    390 mechanical rupture or melting traces in the zone of observation. The obtained microphotographs
    391 showed non-oxide (oxygen unsaturated) structures – metastable solid solution and thin eutectic, by
    392 other words, movable and adherent to the surface films of the first order that corresponds to the
    393 submicron structure of the material under erosion wear with the presence of formation and removal
    394 of brittle secondary structures of the second order"
  15. What do you mean by "secondary structures of the second order" - explain.
  16. All the "chemical analysis" of the results given in L415-434 must be supported by the literature or specific materials investigations (which are not given in the manuscript - must be added). Now, the paper does not contain any investigation that allows us to confirm the formation of the listed compounds. 
  17. Conclusions are too general and do not provide any scientific novelty. Should be improved.
  18. Exemplary, do you have any evidence to support your conclusion: " It can promote the formation of the fragile sublayer for easier
    455 mechanical or electrochemical its removal for the products where the recast layer is undesired or, in
    456 contrary, strength sublayer to improve wear resistance of the responsible working surfaces of the
    457 product. "
  19. References - in my opinion, should be reduced. In the manuscript, there are many multi-citations likewise [80-85] which should be limited and only the most adequate references should be cited..